
# GNNWR: An Open-Source Package of Spatiotemporal Intelligent Regression Methods for Modeling Spatial and Temporal Non-Stationarity

Ziyu Yin[1], Jiale Ding[1], Yi Liu[1], Ruoxu Wang[1], Yige Wang[1], Yijun Chen[1], Jin Qi[1], Sensen Wu[1], and Zhenhong Du[1]

[1]Zhejiang University, School of Earth Sciences

**Correspondence:** Sensen Wu (wesensengis@zju.edu.cn)

**Abstract.** Spatiotemporal regression is a crucial method in geography for discerning spatiotemporal non-stationarity in geographical relationships, which has found widespread application across diverse research domains. This study implements two innovative spatiotemporal intelligent regression models, namely geographically neural network weighted regression (GNNWR) and geographically and temporally neural network weighted regression (GTNNWR), integrating the spatiotemporal weighted framework and neural networks. Demonstrating superior accuracy and generalization capabilities in large-scale data environments compared to traditional methods, these models have emerged as prominent tools. To facilitate the seamless application of GNNWR and GTNNWR in addressing spatiotemporal non-stationary processes, a Python-based package, GNNWR, has been developed. This article details the implementation of these models and introduces the GNNWR package, enabling users to efficiently apply these cutting-edge techniques. Validation of the package is conducted through two case studies. The first case involves the verification of GNNWR using air quality data from China, while the second employs offshore dissolved silicate concentration data from Zhejiang Province to validate GTNNWR. The results of the case studies underscore the effectiveness of the GNNWR package, yielding outcomes of notable accuracy. This contribution anticipates a significant role for the developed package in supporting future research that leverages big data and spatiotemporal regression techniques.

## 1 Introduction

Spatiotemporal non-stationarity, denoting variations in geographical elements or structures across different temporal and spatial contexts, constitutes an intrinsic attribute of nearly all kinds of geographical processes and phenomena. Geographically Weighted Regression (GWR), a classic methodology for delineating spatial non-stationarity in geographical relationships, facilitates the variations of parameter coefficients within the regression equation according to spatial locations (Brunsdon et al., 1996). As a foundational algorithm within the domain of spatiotemporal regression analysis, GWR has been widely used across diverse research domains, including environmental studies (Yang et al., 2019; Shen et al., 2023), urban studies (Sisman and Aydinoglu, 2022; He et al., 2023), and the social sciences (Stein et al., 2015; Lewandowska-Gwarda, 2018; Ahadnejad Reveshty et al., 2023).





On the basis of GWR, various methods have been proposed that focus on optimizing the model ability to solve
spatiotemporal non-stationary relationships. The improvements mainly include the following aspects: the selection
of spatiotemporal distance metrics (Fotheringham et al., 2015; Lu et al., 2014), the choice of weight kernel functions
(Fotheringham et al., 2017), and the optimization of statistical diagnostic methods (Brunsdon et al., 1999; Leung
et al., 2000). Notably, multi-scale geographically weighted regression (MGWR) extends the weight kernel function
to possess varying bandwidths for each independent variable and further enhances the model capacity to fit spatial
non-stationarity (Fotheringham et al., 2017). To deploy the MGWR model, researchers developed a Python-based
(van Rossum, 2011) software package MGWR that focuses on multi-scale estimation and efficient computation of
spatial non-stationarity (Oshan et al., 2019). It supplements R-language-based (R Core Team, 2017) open-source
tools (e.g., spgwr (Bivand and Yu, 2023), gwrr (Wheeler, 2022), and GWmodel (Lu et al., 2024)), improving the
overall accessibility of GWR and MGWR methods.

Owing to the intricate linear interplay between spatial distance and non-stationary weights inherent in geograph-
ical processes, the precise computation of the weight matrix through simple kernel functions encounters notable
challenges. In response to this, diverse methodologies within the domain of geospatial artificial intelligence (GeoAI)
have been proposed to effectively capture the non-linear spatial relationships among pertinent factors (Georganos
and Kalogirou, 2022; Hagenauer and Helbich, 2022). The majority of existing GeoAI approaches utilize neural net-
works in an opaque manner for establishing spatial relationships, leading to a constrained spatial interpretability of
the estimated relationships. To address this, researchers have integrated a spatiotemporal weighted framework with
neural networks, leading to the formulation of spatiotemporal intelligent regression models. Notably, the geograph-
ically neural network weighted regression (GNNWR) model has been introduced, which employs neural networks
to learn the non-linear relationship between spatial distance and non-stationary weights (Du et al., 2020a). Taking
inspiration from GWR, GNNWR employs a spatially weighted neural network (SWNN) to accurately derive the
spatial weight matrix. Subsequently, this SWNN is combined with an ordinary linear regression (OLR) model to
estimate spatial non-stationarity.

In addition to space, time is another fundamental dimension associated with geographical processes. In recent years,
numerous studies have focused on incorporating temporal effects into GWR model to account for both temporal and
spatial non-stationarity (Huang et al., 2010; Fotheringham et al., 2015). Recognizing that time and space exhibit
distinct scale effects, Huang et al. (2010) proposed a straightforward approach to combine spatial and temporal
distances into a unified space-time distance, leading to the development of the geographically and temporally weighted
regression (GTWR) model. The GTWR model, along with its extended methodologies, has been effectively applied
across various domains, producing remarkable results and offering satisfactory interpretability (Ma et al., 2018; He
and Huang, 2018; Guo et al., 2021; Wang et al., 2022).

However, the form of space-time distance usually requires a priori assumption and should be assumed to be
relatively simple (e.g., linear weighted function) so as to eliminate the estimation problem in the terminal model.
Considering that neural networks have the potential to capture the complex non-linear effects in space-time, Wu et al.





(2021) proposed a spatiotemporal proximity neural network (STPNN) to accurately generate space-time distance and extended GNNWR with the STPNN to incorporate temporal effects into spatial non-stationarity. Accordingly, a spatiotemporal intelligent regression model named geographically and temporally neural network weighted regression (GTNNWR) was developed to estimate spatiotemporal non-stationary relationships.

In recent years, GNNWR and GTNNWR have been widely applied in various fields and have achieved excellent fitting capabilities and geographical interpretability, such as atmospheric pollution (Chen et al., 2021; Ni et al., 2022; Liu et al., 2023), environmental modeling (Wu et al., 2020; Du et al., 2021; Wu et al., 2022; Qi et al., 2023) and urban geography (Wang et al., 2022; Yang et al., 2022; Liang et al., 2023). To date, there has been a notable absence of dedicated software pertaining specifically to the GNNWR and GTNNWR models. To facilitate the utilization of these spatiotemporal intelligent regression models by researchers across diverse domains, and to solicit feedback for refinement and enhancement of the models, we have introduced an open-source Python package, denoted as the GNNWR package. This package is designed to furnish a suite of spatiotemporal intelligent regression models, encompassing the GNNWR and GTNNWR variants, thereby serving as a resource for researchers seeking to address challenges within their respective fields.

The GNNWR package offers a comprehensive workflow analysis capability, enabling users to create datasets, instantiate models, conduct training, and generate output results, as well as perform model predictions and visualizations. The GNNWR package uses PyTorch as a deep learning framework (PyTorch Team, 2019), and its dynamic computational graph makes model construction and debugging more intuitive. This package provides extended models as well as great flexibility, allowing advanced users to design custom models based on existing models using the PyTorch framework.

## 2  Model Review

This section offers a concise overview of the GNNWR family of models, which are accommodated by the GNNWR package. The regression formula of the classic OLR model is expressed as:

$$y_i = \beta_0 + \sum_{k=1}^{p} \beta_k x_{ik} + \varepsilon_i \quad \text{for } i = 1, 2, \ldots, n \tag{1}$$

where $y_i$ and $x_{i1}, x_{i2}, \ldots, x_{ip}$ are the dependent variable and independent variables; $\beta_0, \beta_1, \ldots, \beta_p$ are the regressive coefficients, and $\varepsilon_i$ is the error term. The GWR model extends OLR approach to enable spatially localized estimates, the formula is represented as:

$$y_i = \beta_0(u_i, v_i) + \sum_{k=1}^{p} \beta_k(u_i, v_i) x_{ik} + \epsilon_i \quad \text{for } i = 1, 2, \ldots, n \tag{2}$$

Based on the geographically weighted idea similar to GWR, the GNNWR model considers the spatial non-stationarity of the regression relationship as the fluctuation change as the fluctuation change at different locations on the OLR





levels (Du et al., 2020a). In the GNNWR model, the spatial non-stationarity can be represented as:

$$y_i = w_0(u_i, v_i) \times \beta_0^{OLR} + \sum_{k=1}^{p} w_k(u_i, v_i) \times \beta_k^{OLR} x_{ik} + \epsilon_i \quad \text{for } i = 1, 2, \ldots, n \tag{3}$$

where $w_k(u_i, v_i)$ represents the non-stationary weight of the OLR coefficient $\beta_k^{OLR}$. Therefore, the calculation formula for $\hat{y}_i$ is:

$$\hat{y}_i = \sum_{k=0}^{p} \hat{\beta}_k(u_i, v_i) x_{ik} = \sum_{k=0}^{p} w_k(u_i, v_i) \times \hat{\beta}_{k\text{OLR}} x_{ik} \tag{4}$$

The modeling capability of GWR for geographical relationships depends on the fitting and solving ability of the weight kernel function for spatial non-stationarity. The kernel functions of GWR can be abstracted as follows:

$$w_{ij} = f_{\text{kernel}}(d_{ij}, b) \tag{5}$$

Where $d_{ij}$ represents the distance between samples, and $b$ represents the bandwidth.

To accurately fit the complex relationship between spatial proximity and non-stationary weights, the GNNWR model designed a SWNN to realize the neural network expression of the weight kernel function (Figure 1). The spatial weight estimation for point $i$ is calculated as follows:

$$W_i = W(u_i, v_i) = SWNN([d_{i1}^S, d_{i2}^S, \ldots, d_{in}^S]^T) \tag{6}$$

where $[d_{i1}^S, d_{i2}^S, \ldots, d_{in}^S]$ is the distances from location $i$ to training samples.

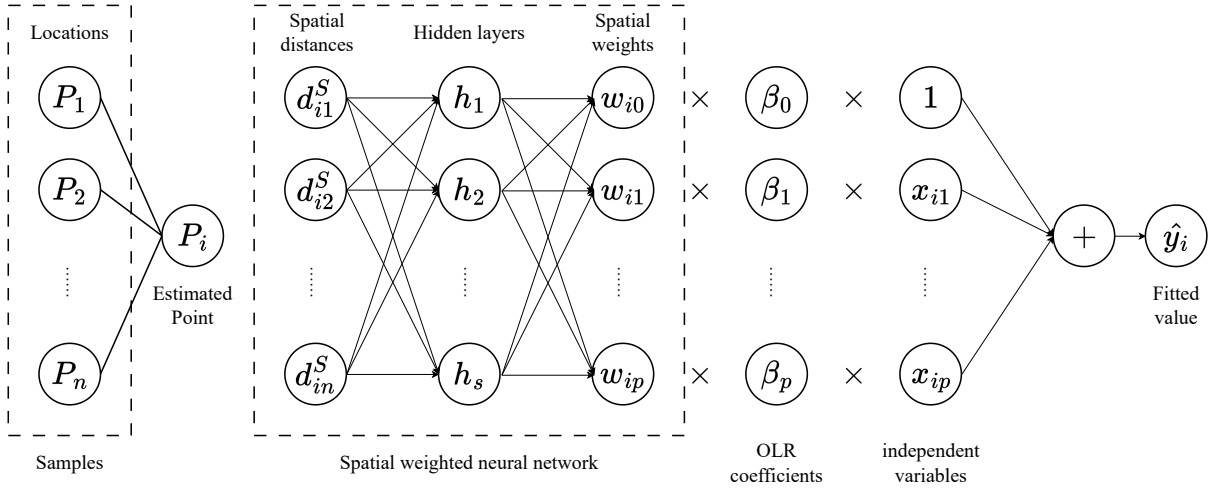

**Figure 1.** The framework of the GNNWR model.





Alongside space, time constitutes a fundamental dimension in the study of geographic phenomena. The GTNNWR model extends the spatial form of the non-stationary relationship in Eq. (3) to the following spatiotemporal form:

$$y_i = \beta_0(u_i, v_i, t_i) + \sum_{k=1}^{p} \beta_k(u_i, v_i, t_i)x_{ik} + \varepsilon_i, \quad \text{for } i = 1, 2, \ldots, n$$


$$= w_0(u_i, v_i, t_i) \times \beta_{0_{OLR}} + \sum_{k=1}^{p} w_k(u_i, v_i, t_i) \times \beta_{k_{OLR}} x_{ik} + \varepsilon_i$$

(7)

where $w_k(u_i, v_i, t_i)$ represents the spatiotemporal non-stationary weight of $\beta_k^{OLR}$, which is determined by its spatiotemporal location $(u_i, v_i, t_i)$ and influenced by other samples. Similar to the SWNN of GNNWR, the GTNNWR model, as shown in figure 2, designed a spatiotemporal weighted neural network (STWNN) calculate the spatiotemporal weights as follows:

$$W(u_i, v_i, t_i) = \text{STWNN}\left(\left[d_{i1}^{ST}, d_{i2}^{ST}, \ldots, d_{in}^{ST}\right]^T\right)$$

(8)

where $\left[d_{i1}^{ST}, d_{i2}^{ST}, \ldots, d_{in}^{ST}\right]$ are the spatiotemporal distances from point $i$ to training samples. This expression indicates that the spatiotemporal non-stationary weight is determined by the spatiotemporal distance. To quantify the spatiotemporal distance, Huang et al. (2010) defined the distance as the following form:

$$d_{ij}^{ST} = d_{ij}^{S} \otimes d_{ij}^{T}$$

(9)

where $d_{ij}^{S}$, $d_{ij}^{T}$, and $d_{ij}^{ST}$ respectively denote the spatial distance, temporal distance, and spatiotemporal distance, and the symbol $\otimes$ represents the operator. To fully capture the nonlinear effects in spatiotemporal, Wu et al. (2021) proposed an spatiotemporal proximity neural network(STPNN) based on the spatial distance $d_{ij}^{S}$ and temporal distance $d_{ij}^{T}$ to generate the spatiotemporal distance $d_{ij}^{ST}$:

$$d_{ij}^{ST} = \text{STPNN}\left(d_{ij}^{S}, d_{ij}^{T}\right)$$

(10)

where $d_{ij}^{ST}$ is a vector that indicates the spatiotemporal proximity between points $i$ and $j$. Therefore, Equation (6) can be refined as:

$$W(u_i, v_i, t_i) = \text{STWNN}\left(\left[d_{i1}^{ST}, d_{i2}^{ST}, \ldots, d_{in}^{ST}\right]^T\right)$$
$$= \text{STWNN}\left(\left[\text{STPNN}\left(d_{i1}^{S}, d_{i1}^{T}\right), \ldots, \text{STPNN}\left(d_{in}^{S}, d_{in}^{T}\right)\right]^T\right)$$

(11)

Therefore, the spatialtemporal weight matrix for any given point across time and space can be derived by merging the STPNN with the STWNN through Eq. (11) (Figure 2). Through integrating global OLR estimates and the

space-time weight matrix, the continuous spatial-temporal variable coefficients is attained. Just like the GTWR model, we can use these coefficients to explain the estimated relationship and make spatiotemporal inference. Hence, the fitted values $\hat{y} = (\hat{y}_1, \hat{y}_2, \ldots, \hat{y}_n)$ are computed as:





$$\hat{y} = \begin{bmatrix} \hat{y}_1 \\ \hat{y}_2 \\ \vdots \\ \hat{y}_n \end{bmatrix} = \begin{bmatrix} x_1^T W(u_1,v_1,t_1)(X^T X)^{-1} X^T \\ x_2^T W(u_2,v_2,t_2)(X^T X)^{-1} X^T \\ \vdots \\ x_n^T W(u_n,v_n,t_n)(X^T X)^{-1} X^T \end{bmatrix} y = Sy \tag{12}$$

where $S$ is the hat matrix of the GTNNWR model.

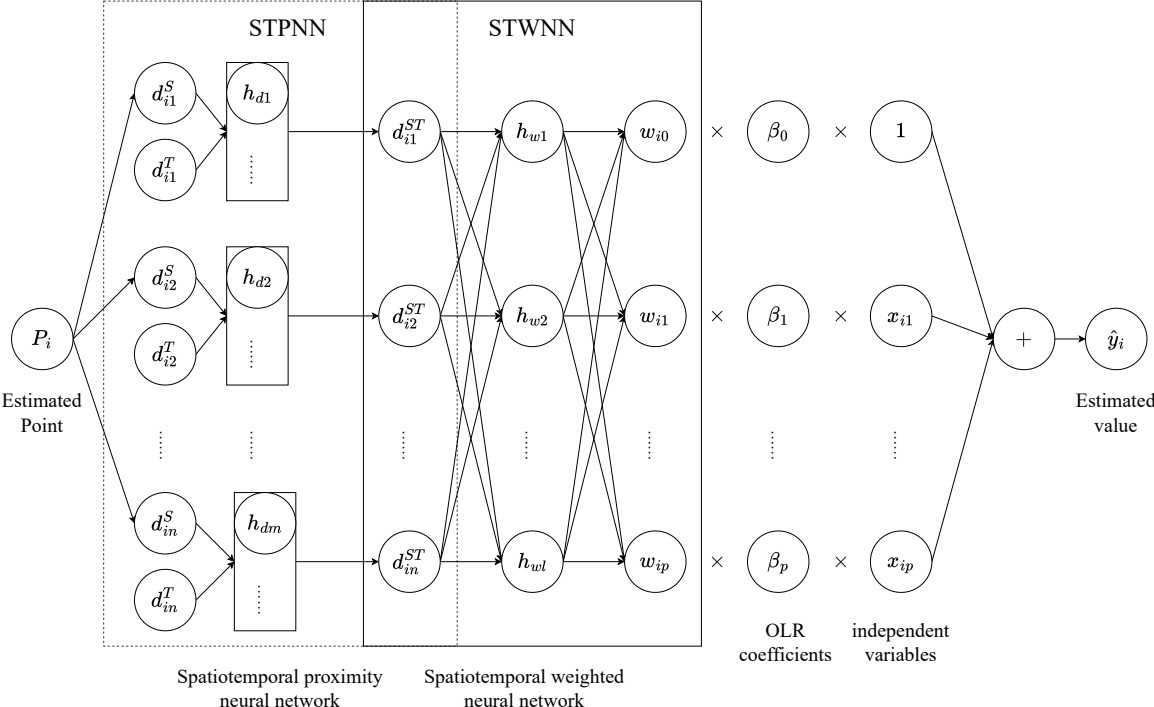

**Figure 2.** The framework of the GTNNWR model.

# 3   Usage Example

In this section, we present a comprehensive overview of the GNNWR package and the range of models it supports. We begin by introducing the fundamental architecture of the software package, delving into its essential components and functionalities. Following this, we outline the analysis process employed in utilizing the package, showcasing its practical application through two case studies.





## 3.1   Package

The GNNWR package is designed with a modular architecture, enabling the integration of diverse module strategies to facilitate a variety of task workflows. It comprises four primary modules: `Dataset`, `Network`, `Utils`, and `Model`.

### 3.1.1   `Dataset`

The `Dataset` module specifies the data types employed throughout the package. It includes the `BasicDataset`
class for training and the `PredictDataset` class for prediction. This module also offers preprocessing functions that convert Pandas' `DataFrame` data into the necessary formats (McKinney et al., 2010), handling tasks such as normalization and dataset partitioning. Additionally, it provides methods for saving and loading datasets, enabling users to directly work with processed data files and instantiate data objects.

### 3.1.2   `Network`

The `Network` module, extending PyTorch's `nn.Module` class, defines the architectures for models such as `SWNN` and `STPNN`. It allows users with programming expertise to customize new network structures based on existing ones, adapting to their specific research requirements.

### 3.1.3   `Utils`

The `Utils` module contains classes for statistical diagnostics and visualization techniques specific to spatial weighted
regression. These diagnostic classes offer a suite of methods to evaluate model performance, while the visualization classes employ map-based representations to enhance the analysis of spatial data and model outcomes.

### 3.1.4   `Model`

The `Model` module is the cornerstone of the package, providing two classes: `GNNWR` and `GTNNWR`. `GNNWR` acts as the foundational class, with `GTNNWR` being its subclass. These classes encapsulate methods for model training, prediction,
diagnostics, and loading. Users can easily invoke these methods to employ the models for problem analysis and forecasting on unseen data.

## 3.2   GNNWR

We commence our investigation by examining air quality modeling through the analysis of data gathered from Chinese air monitoring stations (Du et al., 2020b). This analysis seeks to delineate the spatially non-stationary
associations between PM2.5(particulate matter with aerodynamic diameter less than 2.5 $\mu$m) concentrations and their environmental determinants. Given the pivotal role of PM2.5 as an indicator of air quality, elucidating its spatial variability is crucial for comprehending the underlying spatial processes and environmental dynamics of atmospheric contamination (Han et al., 2016). The objective of this study is to develop a predictive model for the





annual average PM2.5 concentrations in the study area at a 3 km x 3 km spatial resolution for the year 2017.
The model incorporates meteorological variables such as Aerosol Optical Depth (AOD), temperature (TEMP),
precipitation (TP), wind speed (WS), and wind direction (WD), along with elevation data (DEM).

Upon loading the dataset as a Pandas' `DataFrame`, the `init_dataset` function from the GNNWR package is
utilized to convert it into a suitable format for model input. This function proportionally allocates the dataset into
training, validation, and test subsets, and computes the distance vectors for each sample, which are crucial for both
model training and performance evaluation.

In this context, it is essential to specify the independent variables, dependent variables, and spatial position
variables, which correspond to the `x_column`, `y_column`, and `spatial_column` parameters of the `init_dataset`
function, respectively.

When calculating the distance, the `init_dataset` function, by default, uses Euclidean distance to compute the
spatial distances between feature points. This process generates a spatial distance vector for each point, which
serves as input to the neural network component of the model. To accommodate various research requirements, the
`spatial_fun` parameter enables users to provide a custom method for calculating spatial distances.

To optimize the speed of model training and enhance the precision of model outcomes, the function preprocesses
the independent and dependent variables by default. It typically employs normalization for preprocessing; however,
users have the option to adjust the `process_fun` parameter to utilize standardization instead.

```
>>> from gnnwr.datasets import init_dataset
>>> train_set, val_set, test_set = init_dataset(data=data,
...                                             test_ratio=0.2,
...                                             valid_ratio=0.2,
...                                             x_column=x_column,
...                                             y_column=y_column,
...                                             spatial_column=spatial_column)
```

To continue, we need to create an instance of the `GNNWR` model. After importing the `gnnwr.models` module, we
can do so by invoking the `GNNWR` class. The `dense_layers` parameter allows us to specify the number of hidden
layers in the model's neural network, with each layer consisting of a fully connected layer, a batch normalization
layer, a dropout layer, and an activation function. These hyperparameters are closely linked to the neural network's
architecture, encompassing aspects such as the use of a batch normalization layer, the dropout rate, and the activation
function's type, among others. In this specific example, we have configured a neural network with a hidden layer that
includes three sub-layers, each with 512, 256, and 128 nodes, respectively. The activation function uses a PReLU
function with an initial value of 0.1, while all other settings are kept at their default values.

The `GNNWR` class uses Adadelta as its default optimizer, with an initial learning rate of 0.2, and employs cosine
annealing warm restart as its learning rate adjustment strategy. The class also supports a range of optimizers,





including Stochastic Gradient Descent (SGD), Adam, Adagrad, RMSprop, and various learning rate adjustment strategies, such as multistep and cosine annealing. These optimizers and strategies contribute to improving the
model's training efficiency and performance, thereby enabling it to better accomplish its tasks.

To streamline model training, we can utilize the `run` function to specify the number of iterations and the frequency of printing training process information, allowing us to monitor the training progress and performance. Throughout the training process, we will retain the best-performing model within the validation set to prevent the GNNWR model from overfitting. Selecting the optimal model helps minimize the expected error and guarantees that the
model possesses superior generalization ability. For storage convenience, the model repository will only retain a file containing the neural network components of the model. This file encapsulates the structural configuration and parameter information of the neural network.

```
>>> from gnnwr import models
>>> from torch import nn
>>> gnnwr = models.GNNWR(train_dataset = train_set,
...                      valid_dataset = val_set,
...                      test_dataset = test_set,
...                      dense_layers = [1024, 256, 128],
...                      activate_func = nn.PReLU(init=0.2),
...                      start_lr = 0.1,
...                      optimizer = "Adadelta",
...                      model_name = "GNNWR_PM25")
>>> gnnwr.run(max_epoch = 4000, print_frequency = 500)
```

The GNNWR package uses Tensorboard to record the model training process, including the loss and $R^2$ scores
on the training and validation sets for each epoch, as well as the learning rate and best $R^2$ scores obtained on the validation set. By observing the changes in the model during the training process, targeted adjustments to the training method can be made. To enhance users' comprehension of the model architecture, we have incorporated the `add_graph` function. When utilized, this function enables users to visualize the structure of the model within the "Graphs" section of TensorBoard. This functionality not only clarifies the model's architecture but also facilitates
the prompt identification of issues during model debugging and optimization, thereby substantially improving model performance.

We can obtain the composition and results of the model through the `result` method, including the model structure, optimizer structure, used variables, and the accuracy, complexity, and content of statistical tests performed on the model. Among them, the $R^2$ and RMSE(Root Mean Square Error) indicators summarize the model's fitting ability,
while the AIC and AICc indicators provide a deeper understanding of the model's complexity. The $F_1$, $F_2$, and $F_3$ statistical data are used as sample diagnostic measures(Wu et al., 2020). The first two values indicate the





presence of significant spatiotemporal non-stationarity in the model, while the last value evaluates the significance of spatiotemporal non-stationarity in the regression parameters of each independent variable.

```
>>> gnnwr.result()
```

```
------------------Model Information----------------
     Model Name:          | GNNWR_PM25
     independent variable: | ['dem', 'w10', 'd10', 't2m', 'aod_sat', 'tp']
     dependent variable:   | ['PM2_5']

OLS coefficients:
     x0: 7.20676
     x1: -7.51571
     x2: 0.00989
     x3: 21.75958
x4: 30.29816
     x5: -26.90607
     Intercept: 22.79182

     -------------------Result Information---------------
Test Loss: |                  33.63722
     Test R2  : |                   0.84065
     Train R2 : |                   0.82611
     Valid R2 : |                   0.84560
     RMSE: |                        5.79976
AIC:  |                     1798.68992
     AICc: |                     1791.50793
     F1:   |                        0.19219
     F2:   |                        2.40907
     f3_param_0: |                  10.68157
f3_param_1: |                   1.81385
     f3_param_2: |                   3.81711
     f3_param_3: |                  54.19521
     f3_param_4: |                 382.45236
     f3_param_5: |                 117.74578
f3_param_6: |                  21.48253
```





The empirical results reveal that the model exhibits robust performance in the reconstruction of PM2.5 distributions. Notably, the model achieved $R^2$ scores of 0.826 for the training dataset, 0.846 for the validation dataset, and 0.841 for the test dataset. Additionally, statistical analyses confirm the presence of significant spatial heterogeneity
in PM2.5 concentrations, which the model effectively elucidates.

Owing to the intimate association between model analysis and spatial aspects, the GNNWR furnishes a range of spatial visualization functionalities grounded in the folium. By instantiating the `Visualize` object, we can render various model variables within a spatial context. The `Visualize` object proffers multiple visualization techniques, encompassing the visualization of internal datasets within the model, heatmaps of coefficients, and the visualization
of spatial points. Figure 3 illustrates the spatial distribution of the dependent variable PM2.5 across the dataset. Notably, PM2.5 concentrations are elevated in the North China and Xinjiang regions, in contrast to the relatively lower levels observed in Yunnan and the northern reaches of Inner Mongolia.

```
>>> import gnnwr.utils as utils
>>> visualizer = utils.Visualize(data=gnnwr,lon_lat_columns=['lng','lat'])
>>> visualizer.display_dataset(name='all',y_column='PM2_5')
```

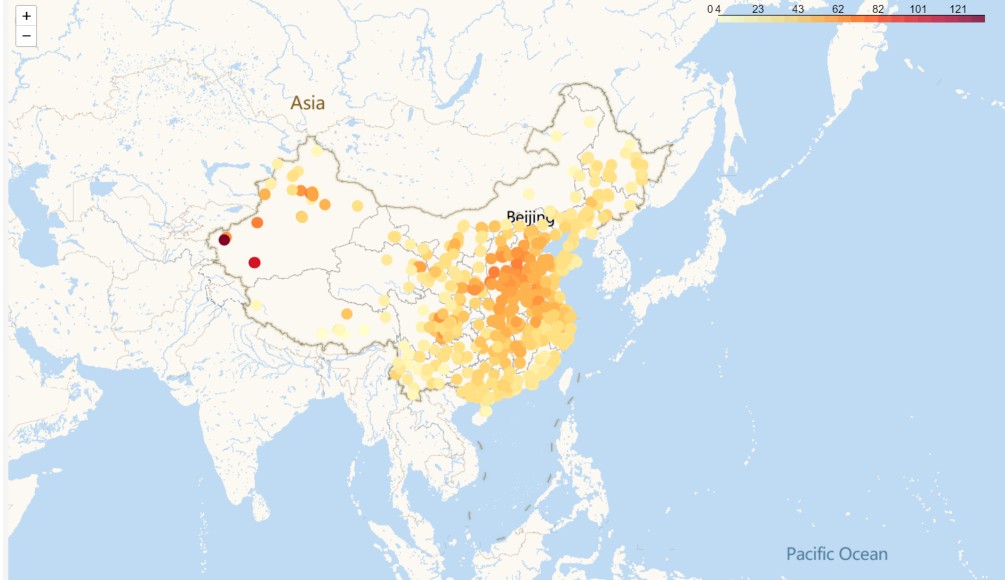

**Figure 3.** Schematic diagram of the spatial distribution of PM2.5

The `coefs_heatmap` function facilitates the visual representation of the spatial distribution of independent variable coefficients, thereby enriching our comprehension of the impact of individual independent variables on the dependent





variable across varying geographical contexts. Figure 4 depicts the distinctive spatial distribution patterns of AOD coefficients.

`>>> visualizer.coefs_heatmap('coef_aod_sat')`

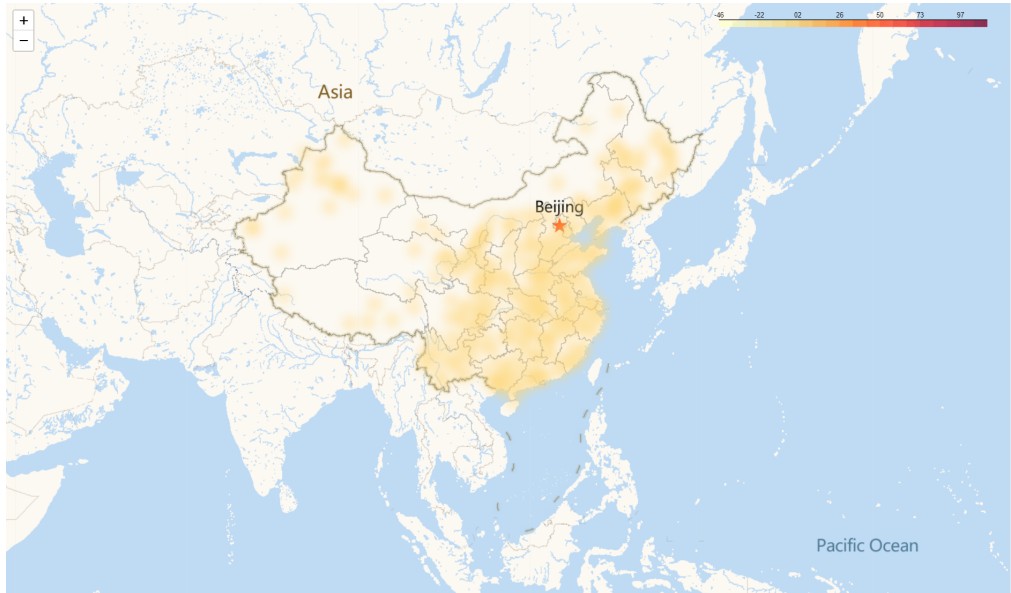

**Figure 4.** Schematic diagram of AOD coefficient distribution.

Through these visualization techniques, we can perceptively comprehend the analysis outcomes of the model. They offer abundant functionality that enables us to better understand the spatial behavior of the model and gain a more profound insight into the model's performance and spatial relationships.

Concurrently, the visualization output of the Visualize object is in HTML format, permitting researchers to
manipulate the map via zooming, panning, and rotation. During the manipulation of the map, the visualization of the data will alter according to the scale of the map. When the map scale is small, the points in the spatial distribution are dense, necessitating the clustering and display of these points to preserve clarity. Conversely, when the map scale is large, the information of the points at specific locations will be displayed. This facilitates detailed inspection and analysis of geographic data to cater to diverse research requirements.

Upon the successful completion of training a model, the frequent need arises to reuse said model. To facilitate this process, the model repository incorporates a dedicated `load_model` function, which is specifically purposed to reload model files that were automatically saved during the training progression. Notably, the repository retains only the neural network-related components within the model. Consequently, when reusing a model, the recommended sequence is as follows: initially, construct an instance correspondent to the model's architectural design, before
subsequently calling the `load_model` method to import the parameters and weights.




Ultimately, we can employ the prediction method to forecast other datasets. Prior to generating predictions, it is essential to transform the other datasets into the `predictDataset` class, which is integrated within the GNNWR package. This transformation can be accomplished by utilizing the `init_predict_dataset` method. This method computes the distance vectors between the features in the dataset to be predicted and the reference points, and

applies the identical scaling transformation to the independent variables as in the training dataset, guaranteeing the reliability of the prediction outcomes. The prediction method yields a Pandas' DataFrame comprising the original data and the predicted results. Moreover, when employing the GNNWR model for analysis, spatial weights are of paramount importance. These weights signify the spatial variability of the influence of each independent variable on the dependent variable. To acquire spatial weights, the `predict_weight` method can be utilized to output the

pertinent information. Figure 5 presents a geographical visualization of the GNNWR model's predictive outcomes.

```
>>> from gnnwr.datasets import init_predict_dataset
>>> pred_dataset = init_predict_dataset(data = pred_data,
...                                     train_dataset = train_set,
...                                     x_column=x_column,
...                                     spatial_column=spatial_column)
>>> res = gnnwr.predict(pred_dataset)
```

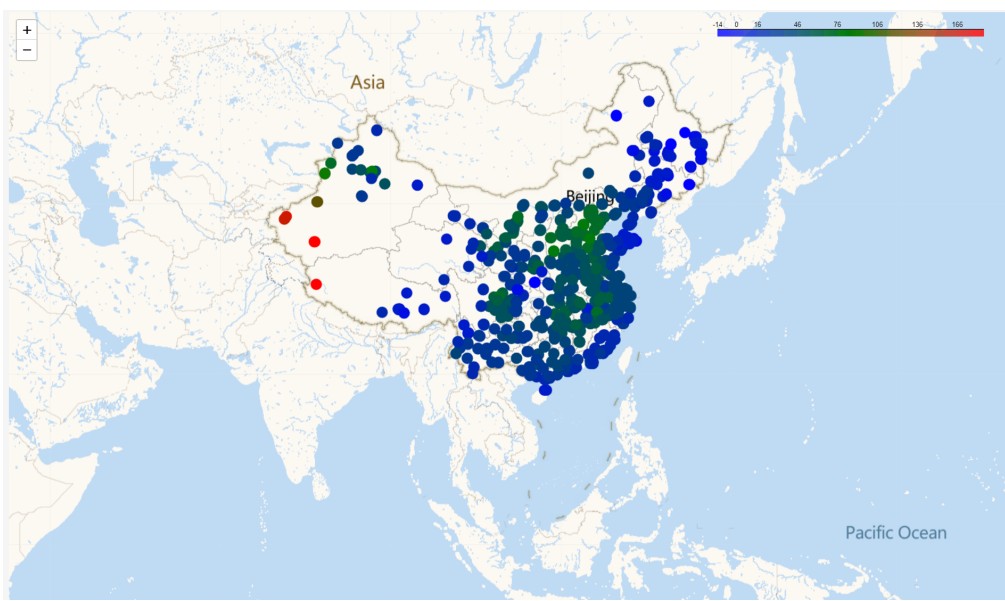

**Figure 5.** Geospatial Visualization of GNNWR Model Predictions for PM2.5





## 3.3 GTNNWR

The workflow of employing GTNNWR is largely akin to that of the GNNWR model. We exemplify this by utilizing daily surface dissolved silicate (DSi) concentration data from the offshore waters of Zhejiang. This study utilized the

GTNNWR approach to retrieve the distribution of coastal DSi concentrations, addressing the challenges posed by spatiotemporal non-stationarity (Qi et al., 2023).

Similar to the GNNWR model, data preprocessing is essential when utilizing the GTNNWR model to acquire a data format that the model can process as well. The GTNNWR model is specifically tailored for spatiotemporal data, wherein the regression coefficients perpetually vary in both space and time. Consequently, the data processed

by the model must possess spatiotemporal attributes. When employing the `init_dataset function`, designating the time data as the time dimension can generate valid input for the GTNNWR model. This function computes the time distance vectors based on the distance calculation method specified by the `temporal_fun` parameter and subsequently employs them as input features for each sampling point. The default time distance calculation method is the Manhattan distance.

```
>>> train_set, val_set, test_set = init_dataset(data=data,
    ...                                 test_ratio=0.15,
    ...                                 valid_ratio=0.1,
    ...                                 x_column=x_column,
    ...                                 y_column=y_column,
...                                 spatial_column=spatial_column,
    ...                                 temp_column=temp_column)
```

`GTNNWR` is designed as a subclass incorporated within the the GNNWR package, inheriting from its foundational `GNNWR` class. As a result, it retains the same set of methods inherent to its superclass. The instantiation process for the GTNNWR model closely mirrors that of GNNWR, with the primary difference lying in the input format

for hidden layers—a two-element two-dimensional list. This unique input configuration stems from GTNNWR's integration strategy, which involves employing a STPNN to compute spatiotemporal proximities and then feeding these computations into a STWNN for determining spatiotemporal weights. Specifically, the first list in this input designates the hidden layer structure of STPNN, whereas the second list delineates the hidden layer architecture pertaining to STWNN.

The GNNWR package features a novel learning rate adjustment mechanism that is employed by default in conjunction with the SGD optimizer. The seminal paper on GTNNWR consistently utilizes this adjustment technique throughout model training (Wu et al., 2021). The process commences with the minimum learning rate, progressively scaling in a stepwise fashion to the maximum learning rate, and then sustaining this peak rate for a designated duration. Thereafter, the learning rate undergoes exponential decay until it reaches a predetermined threshold, be-

yond which it is maintained at a lower fixed value . The adjustment mechanism's hyperparameters encompass the





minimum and maximum learning rates, the decay rate, as well as the number of epochs dedicated to the growth, stabilization, and the decay phase.

The procedure for training an instantiated model with data, as well as the tasks of printing model metadata and exhibiting the outcomes of training, aligns with the methodologies employed in the previous exemplar.

```
>>> gtnnwr = models.GTNNWR(train_set, val_set, test_set)
     >>> gtnnwr.run(max_epoch = 5000,early_stop=200,print_frequency = 200)
     >>> gtnnwr.result()

     -------------------Model Information----------------
     Model Name:            | GTNNWR_DSi
independent variable:  | ['refl_b01', 'refl_b02', 'refl_b03',
                              'refl_b04', 'refl_b05', 'refl_b07']
     dependent variable:    | ['SiO3']
     OLS coefficients:
     x0: 6.84114
x1: 1.63606
     x2: 0.11273
     x3: -5.76276
     x4: 1.62136
     x5: -2.69205
Intercept: 1.05858
     -------------------Result Information---------------
     Test Loss: |                   0.16295
     Test R2  : |                   0.69155
     Train R2 : |                   0.75372
Valid R2 : |                   0.81317
     RMSE: |                        0.40367
     AIC:  |                      453.26982
     AICc: |                      453.26016
     F1:   |                        0.19419
F2:   |                       -8.26737
     f3_param_0: |                 27.48529
     f3_param_1: |                  0.12909
     f3_param_2: |                  0.95790
     f3_param_3: |                  2.41443
```



```
f3_param_4: |                      18.00247
     f3_param_5: |                      28.71637
     f3_param_6: |                     271.65698
```

According to various model indicators, utilizing neural networks to estimate the spatiotemporal non-stationarity of DSi is indeed effective. The model successfully achieved an $R^2$ value of 0.692 on the test set, effectively reconstructing
the distribution of silicate in the offshore waters of Zhejiang. In order to investigate the variability of coefficients for distinct variables across different samples, one can leverage the `reg_result` function to achieve this goal. The function computes and systematically arranges each sample's coefficient values into a Pandas' `DataFrame` format, thereby outputting the results. With the resultant coefficient matrix in hand, researchers can then conduct targeted analyses pertinent to specific spatial processes. For instance, we can visualize the coefficients of each variable by
employing time and space as three-dimensional coordinate axes (Figure 6). This enables us to directly observe the relationship between the bands of remote sensing images and silicate concentrations at different spatial locations. By integrating relevant prior knowledge, we can interpret the outcomes of the model.

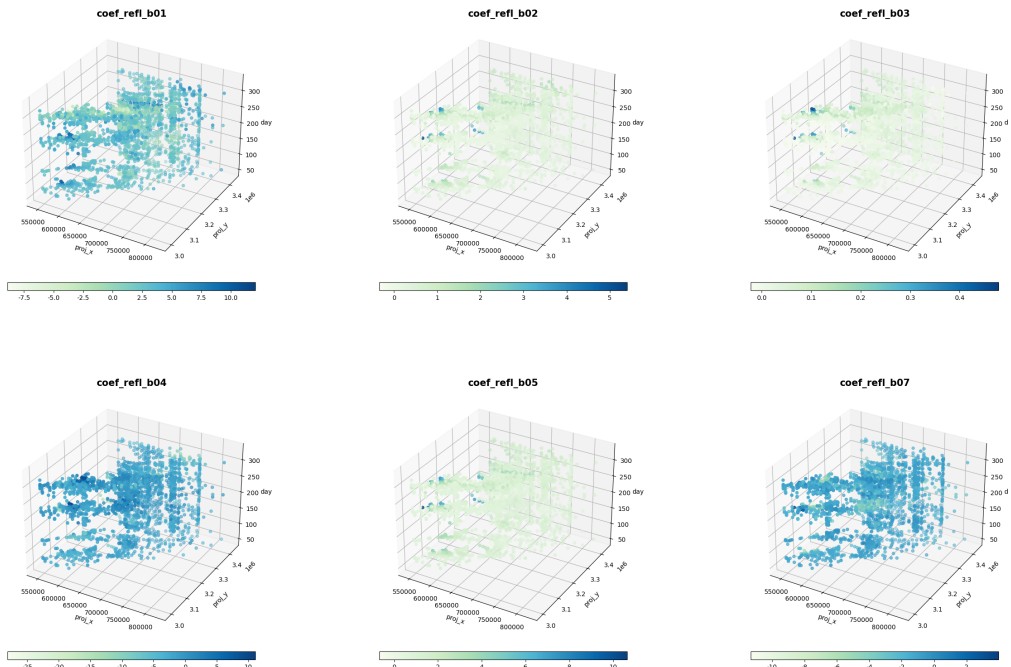

**Figure 6.** Visualization results of model coefficients



## 4    Conclusions

This study introduces the GNNWR package, a Python-based model repository designed to facilitate spatiotemporal
intelligent regression modeling. The package is constructed on PyTorch, a widely employed deep learning frame-
work, and affords a comprehensive workflow for simulating geographical processes characterized by spatiotemporal
non-stationarity. The GNNWR package optimizes intricate procedures encompassing data preprocessing, network
architecture formulation, model training, and result computation, thereby enhancing user accessibility. It enables in-
dividuals with limited programming expertise to quickly master the application of pertinent models such as GNNWR
and GTNNWR for the estimation of spatiotemporal non-stationary processes.

The GNNWR model family exhibits substantial promise in addressing spatiotemporal non-stationarity, effectively
capturing complex nonlinear relationships inherent in the interplay between spatiotemporal proximity and non-
stationary weights. Employing advanced neural network techniques, these models enhance the precision of discerning
spatiotemporal non-stationary features. Additionally, the exploration of spatiotemporal non-stationarity facilitates an
in-depth examination of spatial analytical patterns and the underlying mechanisms governing geographical processes.
The GNNWR and GTNNWR models empower researchers to derive more accurate and reliable insights across diverse
domains within the context of geographical information.

The scholarly understanding of spatiotemporal non-stationarity is progressing, driving the continual evolution of
GNNWR-based models and the emergence of diverse derivatives, such as geographically convolutional neural network
weighted regression (Dai et al., 2022) and directional geographically weighted neural network Regression (Wu et al.,
2019). These model variants have substantially augmented the functionalities of GNNWR across various dimensions.
Moving forward, we are committed to enhancing the model library by leveraging the current framework and inte-
grating a variety of network and data architectures to create novel extension models. This expansion will enhance
the package's capability to incorporate a wide range of modeling techniques for addressing spatiotemporal non-
stationarity. Consequently, this broadening of capabilities will extend the applicability of the models, encompassing
a more comprehensive array of spatiotemporal analytical approaches.

*Code and data availability.* The GNNWR package version used in this article is 0.1.11, which can be found at https://pypi.
python.org/pypi/gnnwr. The project's hosting and development are both ongoing on https://github.com/zjuwss/gnnwr (Yin
et al., 2024a)(https://doi.org/10.5281/zenodo.10890176), and the relevant documents can be found at https://gnnwr.github.
io.All the examples mentioned in this article, supported by research papers, may be retrieved from Yin et al. (2024b)(https:
//doi.org/10.5281/zenodo.10890255). We strongly encourage readers to replicate, adapt, and undertake additional experiments
using this open-source package.



*Author contributions.* YZY, DJL, LY, QJ, and WSS initially developed the package and spearheaded its subsequent evolution. Notably, substantial code enhancements were made by YZY, DJL, LY, WRX, WYG, QJ, CYJ, WSS, and DZH. Each author

430 actively participated in the design discourse and offered critical feedback on the evolving codebase. The manuscript was primarily composed by YZY, WRX, and WYG, with substantive input from all co-authors.

*Competing interests.* The authors declare that they have no conflict of interest.

*Acknowledgements.* This work was supported by the National Natural Science Foundation of China (grant 42271466, 42001323, 423B1001), National Key Research and Development Program of China (grant 2021YFB3900902), Provincial Key R&D

435 Program of Zhejiang (grant 2021C01031), Fundamental Research Funds for the Central Universities (grant 2022FZZX01-05). This work was also supported by the Deep-time Digital Earth (DDE) Big Science Program and the Earth System Big Data Platform of the School of Earth Sciences, Zhejiang University.



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
