# Peer review of "GNNWR: An Open-Source Package of Spatiotemporal Intelligent Regression Methods for Modeling Spatial and Temporal Non-Stationarity"

_Geoscientific Model Development, 2024_

## Referee Comment (RC3)

**Review of the manuscript:**

**GNNWR: An Open-Source Package of Spatiotemporal Intelligent Regression Methods for Modeling Spatial and Temporal Non-Stationarity**

This work introduces a software package called GNNWR for spatiotemporal regression tasks using neural networks. The package is a following implementation of the works of Du et al. (2020a) and Wu et al. (2020). The work further validates the implementation on two works namely, Du et al. (2020b) and Qi et al. (2023).

**General comments**

In my opinion, the manuscript lacks novelty and scientific contribution since it describes a typical framework which is used in most papers/works using machine learning. In addition, there are many similar frameworks such as Pytorch Lightning by Falcon et al. (2020) and there was no discussion about other existing frameworks. The model also seems very similar to Graph Neural Network (GNN) i.e., see Scarselli et al. (2009).

How does the proposed software differ from other deep learning frameworks such as Pytorch Lightning by Falcon et al. (2020)?

Regarding the model:

How does GNNWR differ from the idea of Graph Neural Network (GNN)?

Was there a comparison between GNNWR and other baselines which are widely used as regression models such as Multi-Layer Perceptron (MLP), XGBoosts, and GNN?

I think a discussion about a critical limitation is missing in the manuscript i.e., the estimation of each point is dependent on the distances to all points in the dataset. How is this scaled with large-scale data or global domain? And why shouldn't the estimation be dependent on the neighborhood? Is it essential to have an estimation based on all points in the dataset?

Since the spatiotemporal weights $w_{ip}$ are computed by a neural network (hidden layers), I suspect that $\beta_p$ can be removed because the network can learn these weights when estimating $w_{ip}$. Can you please comment on this?

**Specific and technical comments**

**Lines 5-7:** Please rephrase this sentence to make it clearer.

**Line 75:** Please update the reference for PyTorch i.e.:

Paszke, A., Gross, S., Massa, F., Lerer, A., Bradbury, J., Chanan, G., ... & Chintala, S. (2019). Pytorch: An imperative style, high-performance deep learning library. *Advances in neural information processing systems*, *32*.

**Equation (2):** What are $u_i$ and $v_i$? Do you mean the coordinates, please introduce the terms.

**Equation (4):** What is $\hat{y}_i$ and where is the bias term? If you denote the estimated value as $\hat{y}_i$, please indicate this when you introduce the term.

**Equation (5):** What is $w_{ij}$ and what are $i$ and $j$?

**Line 97:** What happens if we use an inverse of the distance?

**Figure 1:** What are the hidden layers? Do you mean Multi-Layer Perceptron?

**Line 116:** What is this symbol $\otimes$? Do you mean the Hadamard product?

**Lines 125-126:** "Just like the GTWR model, we can use these coefficients to explain the estimated relationship and make spatiotemporal inference." Was there an example of using the coefficients to explain the estimation described in the manuscript? Can you please refer to it?

**Equation (12):** What is $X$ and where is $\beta$?

**Line 130:** Please modify the name of this section. It is about the package description rather than a usage example.

**Lines 181-187:** Is a random split used to generate the training, validation and test sets? That means both validation and test sets will have the same distribution and will be similar to the training set.

**Lines 189-191:** Note that dropout in most deep learning frameworks such as Pytorch scales the output during inference to preserve the distribution inside the neural network. Does dropout have a similar affect for regression as for classification tasks? Was dropout also effective for the regression tasks mentioned in the manuscript?

**Line 239:** What are AIC and AICs?

**Lines 254-265:** Are the reported metrics for the test set?

**Lines 303-306:** "This method computes the distance vectors between the features in the dataset to be predicted and the reference points, and applies the identical scaling transformation to the independent variables as in the training dataset, guaranteeing the reliability of the prediction outcomes." This doesn't guarantee reliability. It guarantees that the input for the model inference follows the same statistical distribution of the training data.

**Figures:** It would be better to provide more description in the captions.

**Line 425:** "… may be retrieved …" What is meant by *may be*?

**Lines 408-409:** "Employing advanced neural network techniques, these models enhance the precision of discerning spatiotemporal non-stationary features." What is exactly meant by advanced neural networks techniques? I do not think MLP is an advanced neural network. Advanced neural networks include i.e., ViT by Dosovitskiy et al. (2020) etc.

**Lines 543-545**: Please correct the date for the reference Wu et al. (2020).

**References:**

Dosovitskiy et al. (2020): https://doi.org/10.48550/arXiv.2010.11929.

Du et al. (2020a): https://doi.org/10.1080/13658816.2019.1707834.

Du et al. (2020b): https://doi.org/10.12082/dqxxkx.2020.190533.

Falcon et al. (2020): https://zenodo.org/doi/10.5281/zenodo.3530844.

Qi et al. (2023): https://doi.org/10.1016/j.scitotenv.2023.163981.

Scarselli et al. (2009): https://doi.org/10.1109/TNN.2008.2005605.

Wu et al. (2020): https://doi.org/10.1080/13658816.2020.1775836.

---

## Author Comment (AC1)

**Responses to #EC**

The overall similarity rate is low, but there are a few paragraphs are exactly same with the following literature, which is not acceptable:
*Geographically and temporally neural network weighted regression for modeling spatiotemporal non-stationary relationships*
The similarity issue has to be thoroughly addressed before resubmission.

**Response**: We thank the editor for his/her kind comment. We feel sorry about the similarity between manuscript and the above-mentioned literature. We have totally rewritten and rephrased the model review section in the revised manuscript. With the revised section, readers will be easier to understand what this article is talking about, and what can the proposed package be used for. Meanwhile, the revised chapters are derived from reorganizing and summarizing references, avoiding high similarity to existing literature.

**Responses to #RC1**

This paper presents an open-source Python package, GNNWR, which incorporates spatiotemporal intelligent regression techniques to model spatial and temporal non-stationarity. The package includes two models: geographically weighted neural network regression (GNNWR) and geographically and temporally weighted neural network regression (GTNNWR). These models utilize neural networks to improve accuracy and generalization over traditional methods, particularly in large-scale data settings. The GNNWR package provides a complete workflow for data preprocessing, model training, and result computation, making advanced spatiotemporal regression techniques more accessible to users.

**Response:** We are very grateful for the reviewer's recognition. We will provide specific answers to each of your questions below.

Part of the content in the 'Model Review' section of the article is somewhat similar to the content found in 'Geographically and Temporally Neural Network Weighted Regression for Modeling Spatiotemporal Non-Stationary Relationships', especially lines 80-85. Please correct.

**Response:** Thanks, the Model Review section offer a brief introduction about the model we have re-implemented in this article. We did have referred to the original article you mentioned, so it might be somewhat similar. In the revised article, we have rearranged the section to avoid direct quotations or similar expressions. For the original lines 80-85 (or the beginning part of Section 2), we have reshaped it. Now it begins with an overall introduction, followed with description subsection for OLR, GWR, GNNWR and GTNNWR:

> *This section offers a concise overview of the GNNWR family of models, which are accommodated by the GNNWR package. Detailed descriptions and performance analysis can be referred to the original articles (Du et al., 2020a; Wu et al., 2021)...*

The article provides a detailed description of the usage workflow for the package within the application case. However, incorporating a clear flowchart to depict the package's operation would greatly benefit the readers.

**Response:** Thank you for your suggestion. We agree with you that a clear flowchart will greatly help readers understand the operation of this software package. We have attached a detailed flowchart in the revised manuscript, which will precisely illustrate the steps and workflow of using this software package.

[Figure]

*Figure 3. Workflow diagram of the package. Dashed boxes denote the raw data, solid boxes represent the code process modules, and arrows indicate the direction of data flow.*

It is recommended to add statistical characteristics of regression coefficients to the model result information. These statistical characteristics provide an overall understanding of the regression coefficients, which is a very important part for users to understand the model results.

**Response:** Thank you for your suggestion. We agree on the importance of incorporating statistical characteristics of the regression coefficients into the model's output information. These details, including standard errors, t-values, p-values, and confidence intervals, will be added to the upcoming version of the package.

In the package, it is mentioned that only the neural network part of the model is saved during the model saving process. It is necessary to introduce to the readers what content is included in this neural network part.

**Response:** Thank you for this comment. When the model is saved, the neural network portion specifically encompasses the network's architecture and associated parameters. We have revised the manuscript and provided a detailed explanation of the components.

*Notably, the repository maintains only the neural network-related components, specifically the neural network architecture and parameters, within the model.*

The article mentions that the "predict_weight" function in the model library can output relevant information. In addition to the spatial weight, all other relevant information should be detailed in the article.

**Response:** Thank you for your suggestion. We have elaborated on all the relevant information generated by the "predict_weight" function in the revised manuscript.

Please review the article for instances where abbreviations are used without being fully expanded. For instance, the abbreviation "PReLU" is encountered for the first time on line 194 without its

full name being provided.

**Response:** Thank you for pointing this out. We have reviewed the article and ensure that all abbreviations are fully expanded upon first mention.

There are several formatting issues within the equations that the authors need to review and correct.

(1) There is a missing 'for i = 1, 2, ..., n' in the second line of Equation 5 on Page 5.

(2) In Equation 11 on Page 5, the equals signs across the two lines are not aligned, whereas in all other equations the equals signs are aligned.

**Response:** Thank you for your attention to the formatting issue. We have reviewed and corrected the formula, including adding missing text in Equation 5 and aligning the equal sign in Equation 11. Additionally, other aspects of the article have been carefully checked to ensure there are no further errors or omissions.

**Responses to #RC2**

This manuscript introduces a Python package named GNNWR, which integrates two neural network-based models, GNNWR and GTNNWR, to enhance the accuracy and interpretability of spatiotemporal data analysis. The GNNWR package is validated using case studies involving air quality and offshore dissolved silicate concentration data. The results, evaluated through various performance metrics, demonstrate the package's robustness and efficacy in capturing both spatial and temporal patterns. The comprehensive implementation and validation demonstrate the potential of this package as a valuable tool in the field of spatiotemporal modeling. Still, there are some points that need to be improved:

**Response:** We are very grateful for the positive comments on this work by the reviewer. Specific answers to each of your questions will be provided below.

Since the GNNWR and GTNNWR are models that have been proposed in other articles. This article should focus less on the model review, while highlight the design and usage of the newly developed package.

**Response:** Thank you for your suggestion. Though the GNNWR and GTNNWR model has been proposed and discussed in other article, we still believe that it is important and beneficial to have a brief review in this article, at least for the algorithm itself. Readers will be easier to understand what this article is talking about, and what can the proposed package be used for. Therefore, we still reserve the Model Review section. But we have revised the section to make it brief and understandable.

Chapter 3 is named as "Usage Example", but it contains class design, function usage, and other content that is outside the scope of Usage Example. Therefore, I suggest to use a more appropriate chapter name.

**Response:** Thanks for your kind advice, we have renamed Chapter 3 to a more appropriate title to better reflect its content.

The usage example for GNNWR and GTNNWR in Sections 3.2 and 3.3 can be further divided into subsections for easier reading.

**Response:** Thank you for your suggestion. The usage examples in Sections 3.2 and 3.3 have been divided into subsections to improve readability and organization.

The design of optimizers is mentioned in both the GNNWR and GTNNWR sections. In L197, a range of optional optimizers are indicated, while in L345, the passage in focus on explaining the SGD optimizer. Is the choice of optimizer model-dependent (determined by GNNWR or GTNNWR)? If so, then it should be specified in the article, if not, then it might be better to arrange all the discussion about optimizers in consecutive paragraphs.

**Response:** Thanks, for keeping the article brief, we have removed the detailed discussion about SGD.

The fonts in the subscripts of the formulas need to be checked further, i.e., the "OLR" in formulas

3, 4, and 7 have both roman and italic types.

**Response:** Thank you for pointing this out. We have reviewed and standardized the font types in the subscripts of the formulas to ensure consistency.

In L116, "the symbol $\otimes$ represents the operator", what is "the operator"?

**Response:**    The description has been added to the paragraph. Symbol $\otimes$ represents a fusion operator which integrates temporal and spatial distance into a spatiotemporal distance.

In Figure 2, the abbreviations (STPNN & STWNN) and their full names are duplicated, instead, detailed descriptions about the network (i.e. the ds and dst nodes) should be added.

**Response:** Thank you for your suggestion. We have revised Figure 2 to remove duplicated full name.

[Figure]

**Responses to #RC3**

This work introduces a software package called GNNWR for spatiotemporal regression tasks using neural networks. The package is a following implementation of the works of Du et al. (2020a) and Wu et al. (2020). The work further validates the implementation on two works namely, Du et al. (2020b) and Qi et al. (2023).

General comments

In my opinion, the manuscript lacks novelty and scientific contribution since it describes a typical framework which is used in most papers/works using machine learning. In addition, there are many similar frameworks such as Pytorch Lightning by Falcon et al. (2020) and there was no discussion about other existing frameworks. The model also seems very similar to Graph Neural Network (GNN) i.e., see Scarselli et al. (2009).

**Response:** We appreciate your feedback on our manuscript. First of all, while it is true that our framework employs techniques commonly used in machine learning, the novelty of our work lies in the specific application and integration of these techniques to address spatiotemporal non-stationarity in geographical processes. To be more specific, our models, GNNWR and GTNNWR, specifically integrate a spatiotemporal weighted framework with neural networks. This allows us to capture complex, non-linear relationships in both spatial and temporal dimensions, which is a significant advancement over traditional methods. Besides, the open-source Python package we have developed not only implements these models but also provides comprehensive workflow capabilities for data preprocessing, model training, prediction, and visualization. This package is tailored for researchers dealing with spatiotemporal data, making advanced spatiotemporal regression techniques more accessible. Therefore, we believe that this manuscript does have its novelty and scientific contribution. To highlight the contribution and novelty, we have revised the manuscripts, specifically in the introduction section, to describe the specific contribution of this work.

> *This research has developed an open-source Python package, denoted as the GNNWR package, to furnish a suite of spatiotemporal intelligent regression models, encompassing the GNNWR and GTNNWR variants, thereby serving as a resource for researchers seeking to address challenges within their respective fields. The GNNWR package offers a comprehensive workflow analysis capability, enabling users to create datasets, instantiate models, conduct training, and generate output results, as well as perform model predictions and visualizations. The GNNWR package uses PyTorch as a deep learning framework (Paszke et al., 2019), and its dynamic computational graph makes model construction and debugging more intuitive. This package provides extended models as well as great flexibility, allowing advanced users to design custom models based on existing models using the PyTorch framework.*

How does the proposed software differ from other deep learning frameworks such as Pytorch Lightning by Falcon et al. (2020)?

**Response:** The PyTorch Lightning provides a general framework for deep learning models, while our package is an implementation of some particular deep learning models. Besides, the model has some domain-specific features, providing utilities for handling spatiotemporal data.

Regarding the model:

How does GNNWR differ from the idea of Graph Neural Network (GNN)?

**Response:** In our perspective, Graph Neural Network (GNN) is used for learning representations of graph-structured data. It is in essence an embedding process for the struct and node information of the graph. While our work is a combination of neural networks and a weighted regression framework, which allows for localized parameter estimation. This is particularly useful for capturing non-stationary relationships in geographical data, a feature not typically addressed by standard GNNs.

Was there a comparison between GNNWR and other baselines which are widely used as regression models such as Multi-Layer Perceptron (MLP), XGBoosts, and GNN?

**Response:** MLP, XGBoost and GNN are general deep learning models, although they can solve some geographical problems, their interpretabilities are somehow limited. To be more specific, they do not have coefficients for the input variables, while GNNWR can output spatial varying coefficients for the variables, revealing spatiotemporal non-stationarity in geographical processes. Thus, we think it might not be meaningful to compare accuracy with these baselines which lack explainability (Barredo Arrieta, A. et al. Explainable Artificial Intelligence (XAI): Concepts, taxonomies, opportunities and challenges toward responsible AI. Information Fusion 58, 82–115 (2020). https://doi.org/10.1016/j.inffus.2019.12.012). However, we did add comparisons between GNNWR and other traditional, explainable models like GWR and GTWR in the revised manuscripts.

I think a discussion about a critical limitation is missing in the manuscript i.e., the estimation of each point is dependent on the distances to all points in the dataset. How is this scaled with large-scale data or global domain? And why shouldn't the estimation be dependent on the neighborhood? Is it essential to have an estimation based on all points in the dataset?

**Response:** Thanks for your insightful comment. Frankly speaking, the "neighborhood" is hard to be defined. Since each observation can have a varying size for its neighborhood, the input, as well as the overall structure of the network will be instable. So, a better solution is to input all the data into the model without a pre-defined "neighborhood", and the model itself will automatically find out and concentrate on the neighboring samples during the training and optimizing process. Indeed, such strategy will lead to a significant computing and memory consumption, and we have added a discussion about this limitation in Section 4.

Since the spatiotemporal weights $wip$ are computed by a neural network (hidden layers), I suspect that $\beta p$ can be removed because the network can learn these weights when estimating $wip$. Can you please comment on this?

**Response:** We thank for your constructive comment. We have tried to remove the OLS coefficients $\beta p$ from the framework. And honestly, the model can achieve similar performance in most cases. However, separating $\beta p$ can provides practical benefits. The $\beta p$ act as pre-hoc information, leading to faster convergence and more stable training dynamics, so the model will require less epoch for training; and can have more robust estimates, especially in cases with limited data or high noise.

Specific and technical comments

Lines 5-7: Please rephrase this sentence to make it clearer.

**Response**: Thanks, we have revised the abstract to make it clearer and more understandable.

Line 75: Please update the reference for PyTorch i.e.:

Paszke, A., Gross, S., Massa, F., Lerer, A., Bradbury, J., Chanan, G., ... & Chintala, S. (2019). Pytorch: An imperative style, high-performance deep learning library. Advances in neural information processing systems, 32. 2

**Response:** Thanks, we have updated the reference

Equation (2): What are $ui$ and $vi$? Do you mean the coordinates, please introduce the terms.

Equation (4): What is $\hat{y}i$ and where is the bias term? If you denote the estimated value as $\hat{y}i$, please indicate this when you introduce the term.

Equation (5): What is $wij$ and what are $i$ and $j$?

**Response:** Thank you for your detailed suggestion, we have checked and fixed all the term and equation problems, as well as added extra description about the terms.

Line 97: What happens if we use an inverse of the distance?

**Response:** If we use the inverse of the distance, the form of the kernel function needs to be changed accordingly to ensure that distant samples share a smaller weight against each other. We have fixed our expression, using "distance-decaying kernel function" to point out the usage of the kernel function, and use the most common Gaussian kernel as a representative.

Figure 1: What are the hidden layers? Do you mean Multi-Layer Perceptron?

**Response:** Yes, the SWNN is essentially an MLP that translates inter-sample distance relationships into spatial weights.

Line 116: What is this symbol $\otimes$? Do you mean the Hadamard product?

**Response:** It's a fusion function for spatial and temporal distances, we have added the explanation:

*where symbol ⊗ represents a fusion operator which integrates temporal and spatial distance into a spatiotemporal distance*

Lines 125-126: "Just like the GTWR model, we can use these coefficients to explain the estimated relationship and make spatiotemporal inference." Was there an example of using the coefficients to explain the estimation described in the manuscript? Can you please refer to it?

**Response:** We thank for your constructive comment. The coefficients, like in an OLR-based analysis, can reflect the relationship between the independent and dependent variables. In GNNWR and GTNNWR, each unique observation has its own coefficients, which can be used to explain the relationship at the certain spatial/temporal position. We have modified the manuscript, showing how to use the coefficient to explain the estimation. For example, we have changed the caption of figure 6 and figure 9:

*Figure 6. Diagram of AOD coefficient distribution. Darker areas highlight regions with strong positive correlations, indicating high levels of particulate matter*

*Figure 9. Model coefficients in a 3D space-time coordinate system. Blue dots represent a greater positive effect of the variable on silicate concentration; lighter dots represent a smaller effect.*

Equation (12): What is $X$ and where is $\beta$?

**Response:** We have fixed

Line 130: Please modify the name of this section. It is about the package description rather than a usage example.

**Response:** Thanks, we have modified the name into *Package Descriptions* and rearranged the section.

Lines 181-187: Is a random split used to generate the training, validation and test sets? That means both validation and test sets will have the same distribution and will be similar to the training set.

**Response:** Yes, we have added a note about the division:

*This function randomly divides the dataset into training, validation and testing subsets according to the ratio specified in the input parameters;*

Lines 189-191: Note that dropout in most deep learning frameworks such as Pytorch scales the output during inference to preserve the distribution inside the neural network. Does dropout have a similar affect for regression as for classification tasks? Was dropout also effective for the regression tasks mentioned in the manuscript?

**Response:** Yes, the dropout is effective for our regression model. As a matter of fact, GNNWR and GTNNWR has a drop_out parameter with default value of 0.2. We have added more information about this point in the revised version.

Line 239: What are AIC and AICs?

Lines 254-265: Are the reported metrics for the test set?

**Response:** The Akaike Information Criterion (AIC) and its corrected version (AICc) are measures for model selection, in practice, the model with the lowest AIC or AICc value is considered the best among the compared models. And the reported metrics in Line 254-265 is for the test set.

Lines 303-306: "This method computes the distance vectors between the features in the dataset to be predicted and the reference points, and applies the identical scaling transformation to the independent variables as in the training dataset, guaranteeing the reliability of the prediction outcomes." This doesn't guarantee reliability. It guarantees that the input for the model inference follows the same statistical distribution of the training data.

**Response:** Thanks for your constructive comment, we have revised this paragraph:

*This method computes the distance vectors between the features in the dataset to be predicted and the reference points, and applies the identical scaling transformation to the independent variables as in the training dataset, guaranteeing that the input for the model inference follows the same statistical distribution of the training data.*

Figures: It would be better to provide more description in the captions.

**Response**: We thank for your kindly comment, we have added more description in the captions of the figures.

Line 425: "… may be retrieved …" What is meant by may be?

**Response:** We have changed into "can be"

Lines 408-409: "Employing advanced neural network techniques, these models enhance the precision of discerning spatiotemporal non-stationary features." What is exactly meant by advanced neural networks techniques? I do not think MLP is an advanced neural network. Advanced neural networks include i.e., ViT by Dosovitskiy et al. (2020) etc. 3

**Response:** We have revised the conclusion, modifying the relevant statements.

*This study introduces the GNNWR package, a Python-based model repository designed to facilitate spatiotemporal intelligent regression modeling. The package is constructed on PyTorch, a widely employed deep learning framework, and affords a comprehensive workflow for simulating geographical processes characterized by spatiotemporal non-stationarity. The GNNWR package optimizes intricate procedures encompassing data preprocessing, network architecture formulation, model training, and result computation, thereby enhancing user accessibility. It enables individuals with limited programming expertise to quickly master the application of pertinent models such as GNNWR and GTNNWR for the estimation of spatiotemporal non-stationary processes. The integrated visualization functionalities further augment the package's utility, allowing users to interpret model outcomes and their spatial relationships more effectively.*

Lines 543-545: Please correct the date for the reference Wu et al. (2020).

**Response:** We have fixed

References:

Dosovitskiy et al. (2020): https://doi.org/10.48550/arXiv.2010.11929.

Du et al. (2020a): https://doi.org/10.1080/13658816.2019.1707834.

Du et al. (2020b): https://doi.org/10.12082/dqxxkx.2020.190533.

Falcon et al. (2020): https://zenodo.org/doi/10.5281/zenodo.3530844.

Qi et al. (2023): https://doi.org/10.1016/j.scitotenv.2023.163981.

Scarselli et al. (2009): https://doi.org/10.1109/TNN.2008.2005605.

Wu et al. (2020): https://doi.org/10.1080/13658816.2020.1775836

**Responses to #RC4**

The manuscript presents the development and validation of the GNNWR package, which includes novel spatiotemporal intelligent regression models, namely GNNWR and GTNNWR. These models are designed to handle spatial and temporal non-stationarity using neural networks within a spatiotemporal weighted framework. The paper is well-structured and provides detailed explanations of the models and their applications. However, there are several areas that require clarification and improvement. The abstract is concise and informative, but it would benefit from a brief mention of the key results from the case studies to highlight the effectiveness of the proposed models.

**Response:** We are very thankful for the recognition by the Reviewer. The positive assessment on the importance of this work by the Reviewer is very encouraging and very much appreciated. We have addressed every issue that the Reviewer raised below, and revised the abstract to offer a brief mention of the key results from the case study.

General Comments:

The introduction provides a comprehensive overview of the importance of spatiotemporal regression and the limitations of existing methods. It effectively sets the stage for the introduction of the GNNWR and GTNNWR models. However, the manuscript mentions traditional methods but does not provide a detailed comparative analysis. Please find the general comments below:

Please specify the traditional methods.

**Response:** Thanks for your constructive advice. We have added a specific Section 2.1 in the revised manuscript to specify the traditional methods.

It might be helpful to explicitly state the primary contributions of this work in the introduction to clearly differentiate it from previous research.

**Response**: Thanks for your kind advice, we have revised the introduction with a clear outline of the primary contribution of this work.

> *In recent years, GNNWR and GTNNWR have been widely applied in various fields and have achieved excellent fitting capabilities and geographical interpretability, such as atmospheric pollution (Chen et al., 2021; Ni et al., 2022; Liu et al., 2023), environmental modeling (Wu et al., 2019; Du et al., 2021; Wu et al., 2022; Qi et al., 2023) and urban geography (Wang et al., 2022; Yang et al., 2022; Liang et al., 2023). However, the accessible version for the source code of GNNWR (Du, 2019) and GTNNWR (Wu, 2020) are implemented with TensorFlow 1.x, which is too old to run in the latest hardware environment. And the codes are not highly encapsulated, which makes researchers harder to use and develop the model. Therefore, there is a need to develop a set of model implementations with a newer architecture, simpler usage, and clearer code structure to facilitate the utilization of these spatiotemporal intelligent regression models by researchers in different fields, and to solicit feedbacks for refinement and enhancement of these models.*

*This research has developed an open-source Python package, denoted as the GNNWR package, to furnish a suite of spatiotemporal intelligent regression models, encompassing the GNNWR and GTNNWR variants, thereby serving as a resource for researchers seeking to address challenges within their respective fields. The GNNWR package offers a comprehensive workflow analysis capability, enabling users to create datasets, instantiate models, conduct training, and generate output results, as well as perform model predictions and visualizations. The GNNWR package uses PyTorch as a deep learning framework (Paszke et al., 2019), and its dynamic computational graph makes model construction and debugging more intuitive. This package provides extended models as well as great flexibility, allowing advanced users to design custom models based on existing models using the PyTorch framework.*

A brief explanation of how these weights are derived using the SWNN would enhance understanding.

**Response:** Thanks for your constructive advice. We have added a more detailed explanation in the revised Section 2.2.

*Since a pre-defined kernel function might not accurately estimate the complex, heterogenous geographical processes. The GNNWR model introduces a spatially weighted neural network (SWNN) to represent the nonstationary weight matrix.*

*The spatial weight estimation for point i is calculated as follows:*

$$W(u_i, v_i) = SWNN\left(\left[d_{i1}^S, d_{i2}^S, ..., d_{in}^S\right]^\top\right)$$

*where $\left[d_{i1}^S, d_{i2}^S, ..., d_{in}^S\right]$ is the distances from location i to other training samples, and the weighting matrix $W(u_i, v_i)$ is a diagonal matrix, whose diagnostic elements are the non-stationary weights $w_0(u_i, v_i), w_1(u_i, v_i), ..., w_p(u_i, v_i)$ for the regression.*

For the GTNNWR model, the process of integrating the STPNN and STWNN should be explained with more detail, particularly the way spatiotemporal distances are calculated and used.

**Response:** Thanks for your constructive advice. We have added a more detailed explanation in the revised Chapter 2.3.

*To fully capture the nonlinear effects in the spatiotemporal dimension, \cite{WU:2021} proposed a spatiotemporal proximity neural network(STPNN) as the fusion operator $\otimes$. Therefore, the spatiotemporal weight matrix for any given point across time and space can be derived by merging the STPNN with the STWNN.*

$$W(u_i, v_i, t_i) = STWNN\left(\left[d_{i1}^{ST}, d_{i2}^{ST}, ..., d_{in}^{ST}\right]^T\right)$$

$$= STWNN\left(\left[STPNN(d_{i1}^S, d_{i1}^T), ..., STPNN(d_{in}^S, d_{in}^T)\right]^T\right).$$

The case studies provided (air quality data from China and offshore dissolved silicate concentration data from Zhejiang Province) are appropriate for demonstrating the models'

capabilities. However, the results section should include more detailed comparisons with traditional methods to better highlight the advantages of GNNWR and GTNNWR. Add a table or figure that directly compares the performance metrics (e.g., mean squared error, R-squared) of GNNWR and GTNNWR against traditional methods like OLS, GWR, and other spatiotemporal models. It would be beneficial to include visualizations of the results (e.g., maps or plots) to illustrate the models' performance and the spatial/temporal patterns they capture.

**Response:** Thanks for your constructive advice, we have added 2 violin plots with marked R2 to offer a straightforward comparison between GNNWR, GTNNWR and traditional models like OLS, GWR and GTWR.

Ensure that any mention of parallel processing clearly explains its relevance and application within the context of the research. For instance, explain how parallel processing contributes to efficiency and performance in the GNNWR model or mention in limitations.

**Response:** Thanks for your kind comment. The model we implemented is to some extent parallel, as it computes the regression relationship on all sample points simultaneously without traversing them one by one. However, the existing model lacks the parallelizable ability to compute in parallel on multiple compute nodes (i.e. multiple CUDA GPUs). Though Pytorch has a DataParallel model, we have no plans to release a parallelizable version of the model at this time. Because all the observations are related to each other, and we have not yet thought of a way to rationally partition the dataset to train models in parallel without breaking the original geographic relationships. We have mentioned this limitation in Section 4.

The manuscript does not discuss the computational efficiency of the proposed models. Please provide benchmarks comparing the training and prediction times of GNNWR and GTNNWR with those of traditional models. Discuss strategies for improving efficiency, such as parallel processing.

**Response:** Thanks for your constructive advice. As a deep learning model, GNNWR does require more time than traditional models to fit the given dataset. But the extra time consumption is worthy, compared to the improved performance. Also, we can use CUDA-enabled GPU to accelerate the training process, making the time consumption quite acceptable. We have added discussions about the computational efficiency, as well as strategies for improving efficiency in the revised manuscripts.

*It is noteworthy that, as a deep learning model, GNNWR does require more time than traditional models to fit the given dataset. For the above experiment on PM2.5 dataset, it takes 2000 epochs for GNNWR to optimize the network's parameters and minimize the loss, which is about 3 minutes in a CPU (Intel Core i5-12400) environment. Nevertheless, compared to the advantages in model performance, such a time-consumption is acceptable, especially considering that a CUDA-enabled GPU can further accelerate the process.*

*Future works should focus on optimizing the computational efficiency, implementing parallel processing techniques and optimizing the model architecture are potential methods that can significantly reduce computation times.*

The conclusion summarizes the findings but does not address limitations or future work in detail. Potential limitations of the current models are not discussed. Acknowledge limitations such as computational demands, data requirements, and model interpretability. Discuss how these might be addressed in future work.

**Response:** Thanks for your kind comment. We have rearranged the conclusion in the revised manuscripts, adding a discussion on limitations in computational demands, data requirements, and model interpretability, as well as future work for potential solution.

*This study introduces the GNNWR package, a Python-based model repository designed to facilitate spatiotemporal intelligent regression modeling. The package is constructed on PyTorch, a widely employed deep learning framework, and affords a comprehensive workflow for simulating geographical processes characterized by spatiotemporal non-stationarity. The GNNWR package optimizes intricate procedures encompassing data preprocessing, network architecture formulation, model training, and result computation, thereby enhancing user accessibility. It enables individuals with limited programming expertise to quickly master the application of pertinent models such as GNNWR and GTNNWR for the estimation of spatiotemporal non-stationary processes. The integrated visualization functionalities further augment the package's utility, allowing users to interpret model outcomes and their spatial relationships more effectively.*

*However, the GNNWR package is not without limitations. The GNNWR and GTNNWR models are computationally intensive, particularly for large datasets. Also, training neural networks requires substantial computational resources, which may limit accessibility for some users. Future works should focus on optimizing the computational efficiency, implementing parallel processing techniques and optimizing the model architecture are potential methods that can significantly reduce computation times. Dara handling is another area that can be improved, incorporating techniques for spatiotemporal data augmentation and pre-processing can make the models more robust and applicable to a wider range of datasets.*

*Besides, the scholarly understanding of spatiotemporal non-stationarity is progressing, driving the continual evolution of GNNWR-based models and the emergence of diverse derivatives, such as geographically convolutional neural network weighted regression (Dai et al., 2022) and directional geographically weighted neural network Regression (Wu et al., 2019). These model variants have substantially augmented the functionalities of GNNWR across various dimensions. Moving forward, we are committed to enhancing the model library by leveraging the current framework and integrating a variety of network and data architectures to create novel extension models. This expansion will enhance the package's capability to incorporate a wide range of modeling techniques for addressing spatiotemporal non-stationarity. Consequently, this broadening of capabilities will extend the applicability of the models, encompassing a more comprehensive array of spatiotemporal analytical approaches.*

Specific Comments:

Matrices and vectors in the entire manuscript should be denoted by boldface uppercase letters (e.g., Eq. 12).

**Response:** We have fixed

In Eqs. 1, 7, and 2, 3, please clarify the distinction between and .

**Response:** The review system failed to display the symbols you commented, but I guess you meant the distinction between different betas, and we have fixed this point.

Please review the sentence in lines 87-89 to enhance clarity.

**Response:** We have changed:

*Similarly, the GNNWR model describes spatial non-stationarity through fluctuating changes in the coefficients of OLR at different locations (Du et al., 2020a).*

In line 85, the sentence needs a period after the equation. Please check the whole manuscript (e.g., lines 93, 96).

**Response:** We have fixed

In lines 4, 28, 43, and more, each word of abbreviations should begin with a capital letter.

**Response:** We have fixed

In Eqs. 3, 4, and 6, why do you need to use the symbol "×"?

**Response:** Thanks for your comment, we have removed the useless "×"

In Eq. 1, what do p and n represent?

**Response:** We have added a description of them.

Please check all symbols shown in the manuscript and mentioned in the text (such as $\hat{y}_i$, $w$, i, j, etc.).

**Response:** We have checked

In Eq. 5, what is the range of values for "j"?

**Response:** We have added a description in the updated Eq. 4